## PERSPECTIVE

# Direction from the wanderer: insight into cardiac neural control from single axons within the vagus nerve

Varun Malik [ID]

*Cardiac Arrhythmia & Autonomic Neurosciences Research Laboratory, Faculty of Health and Medical Sciences, The University of Adelaide, Adelaide, South Australia, Australia*

Email: varun.malik@adelaide.edu.au

Handling Editors: Harold Schultz & Kalyanam Shivkumar

The peer review history is available in the Supporting Information section of this article (https://doi.org/10.1113/JP287381#support-information-section).

Cardiac neural control remains an enigma and the reward for 'untangling' the role of the 'wandering' vagus in human health and disease, immense. With each heartbeat, the autonomic nervous system (ANS) precisely controls cardiac function, with autonomic dysfunction either triggering or perpetuating several disorders, particularly arrhythmias (Malik & Shivkumar, 2024). Afferent neurons within the vagus form several, complex, nested feedback loops at multiple neuraxial levels, integrating sympathetic and parasympathetic cardiac control. Under resting conditions, vagal efferent activity predominates, resulting in slower heart rates. Recently, insights from direct vagal nerve recordings during exercise in an ovine model challenge the traditional 'yin' and 'yang' view of the parasympathetic and sympathetic components of the ANS, whereby each works in opposition. Rather, cardiac vagus activity increases in concert with sympathetic tone, supporting the exercising heart (Shanks et al., 2023). Such invasive cardiac vagal recordings were hitherto not feasible in humans.

In this issue of *The Journal of physiology*, Farmer and colleagues extend their pioneering use of vagal nerve microneurography in humans (Ottaviani et al., 2020) to measure activity from single axons that synchronizes with cardiac activity (Farmer et al., 2025). Fifteen healthy human participants (aged 19–59) of either sex underwent percutaneous insertion of a sterile tungsten microelectrode using ultrasound guidance into either left or right vagus (in one participant, both vagi, on separate occasions). It was necessary to determine whether the anatomic location was suitable for percutaneous entry (avoiding the carotid artery). The microelectrode was inserted caudal to the carotid bulb, with minor adjustments penetrating different nerve fascicles. Recordings were simultaneous with electrocardiograms (ECG), respiratory rate and blood pressure, and made at baseline and during slow, deep breathing, for each fascicle position.

Single units (axons) were identified, offline, after digital filtering, using template matching functions. Cardiac and respiratory rhythmicity was determined using cross-correlograms assessing variations in axonal firing frequency, phasic with R wave frequency (ECG) and respiration rate, respectively. Further, functional classification was attempted by assessing axonal firing frequency within the cardiac cycle and spike characteristics (positive, suggestive of myelinated axons, and negative, unmyelinated). Positive spikes with cardiac and expiratory (respiratory) rhythmicity were considered to resemble cardioinhibitory efferent neurons. Negative spikes (possible unmyelinated neurons) that displayed cardiac and expiratory rhythmicity were classified as cardiac efferent neurons. The remainder of the positive spikes (possible myelinated neurons) were classified as afferent neurons, which were then further sub-classified based on the timing of their peak frequency within the cardiac cycle as: (1) low-pressure (volume-regulating) cardiopulmonary baroreceptors with inferences made as to whether they respond to stretch, filling or both, based prior animal work; or (2) high-pressure arterial blood-pressure-regulating baroceptors (thought to have peak firing rates prior to the T wave, with rapid onset).

Single units were isolated from the recordings of 9 of the 15 participants studied. It is unclear how many participants were excluded due to unfavourable anatomy, which is of relevance for future studies. There were no major complications. Self-limited adverse effects were headache, pain and throat discomfort. Overall, 44 patterns resembling axons that displayed cardiac rhythmicity were identified (36 from the right vagus nerve). The use of slow deep breathing increased inclusion of neurons that displayed only minimal cardiac rhythmicity at rest. Seven possible cardio-inhibitory neurons were identified with maximal firing frequencies at the lowest heart rate during respiratory sinus arrhythmia. Negative spikes (possible cardiac efferent neurons) also had higher frequency with lower heart rates, supporting their classification as cardiac efferent neurons.

The authors are to be heartily congratulated on an important study that will drive the field of cardiac neuroscience. There are some important limitations, which are also discussed in the paper. Cardiac rhythmicity may not be present in neurons involved in cardiac control. Second, there are limited axon samples, without data regarding discarded recordings owing to lack of rhythmicity. Nevertheless, the finding of increased firing rates in neurons thought to be efferents correlating with lower heart rates is supportive of their involvement in cardiac control. Afferent classification, whilst attempted, could not be confirmed within this experiment and the descriptions are presumptive. Further work with perturbations of cardiac volume and blood pressure (such as using lower body negative pressure) will be imperative to delineate these further (Malik et al., 2021). Accounting for laterality of vagal cardiac control will be also be important.

This work is expected to make substantial contributions toward understanding cardiac autonomic physiology in humans, particularly afferent, regulatory cardiac control in human health and disease, as illustrated in Fig. 1. Afferent autonomic dysfunction occurs in hypertension, heart failure and, recently, also in patients with atrial fibrillation (Malik et al., 2021). In conclusion, this technique provides an unprecedented opportunity to understand cardiac interoception in health and disease.

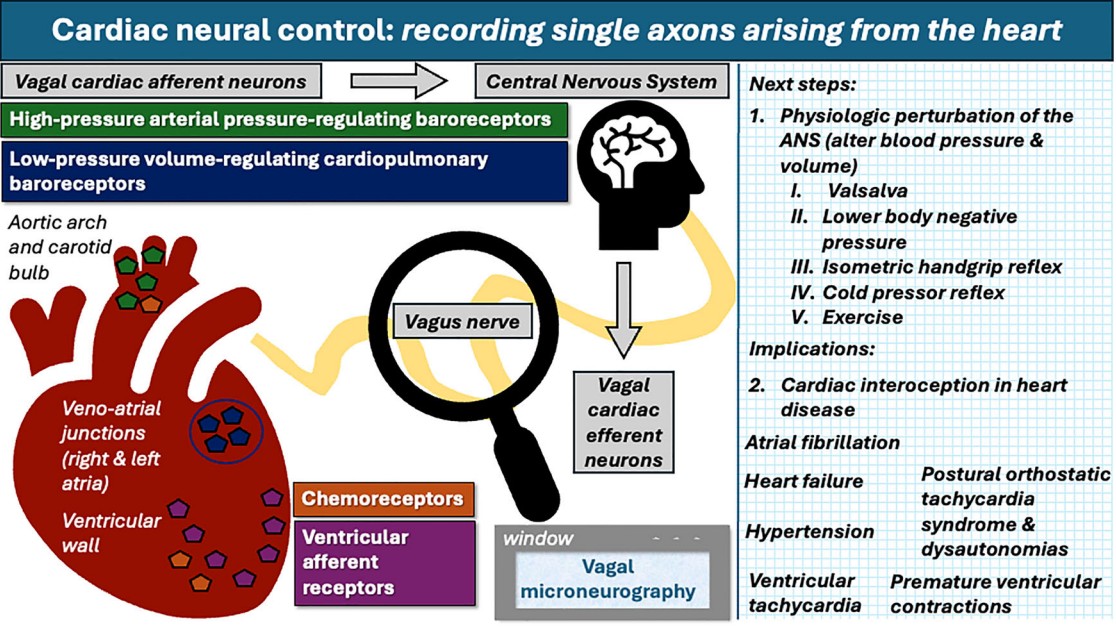

**Figure 1. Vagal microneurography as a 'window' to cardiac neural control**
Vagal cardiac afferent nerves transmit physiological information such as cardiac stretch (recognizing changes in blood pressure and volume). Chemical stressors to the heart are also transmitted via chemoreceptors. This is depicted by the coloured pentagons in the heart (left) and labelled with corresponding-coloured boxes. Cardiac efferent nerves in the vagus arise in the central nervous system and travel to the heart to exert control (chronoptropy, inotropy and dromotropy). On the right is shown proposed next steps in utilizing this novel technique to understand cardiac autonomic physiology in human health and disease as well as implications of vagal microneurography to better understand specific cardiac conditions are shown in the right panel in graph rule. ANS; autonomic nervous system.

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

## Additional information

### Competing interests

The author declares no competing interests.

### Author contributions

Sole author.

### Funding

None.

## Acknowledgements

Open access publishing facilitated by The University of Adelaide, as part of the Wiley - The University of Adelaide agreement via the Council of Australian University Librarians.

## Keywords

afferent neurons, autonomic nervous system, cardiac, efferent neurons, microneurography, respiratory, vagus nerve

## Supporting information

Additional supporting information can be found online in the Supporting Information section at the end of the HTML view of the article. Supporting information files available:

**Peer Review History**

