## [Peer Review History · The Journal of Physiology]

Direction From The Wanderer: Insight Into Cardiac Neural Control From Single Axons Within the Vagus Nerve

Varun Malik

DOI: 10.1113/JP287381

Corresponding author(s): Varun Malik (varun.malik@adelaide.edu.au)

Review Timeline:

Submission Date:	12-Aug-2024
Editorial Decision:	19-Aug-2024
Revision Received:	27-Aug-2024
Editorial Decision:	06-Sep-2024
Revision Received:	11-Sep-2024
Accepted:	12-Sep-2024

Senior Editor: Harold Schultz

Reviewing Editor: Harold Schultz

Transaction Report:

Dear Dr Malik,

Re: JP-P-2024-287381 "Direction From The Wanderer: Insight Into Cardiac Neural Control From Single Axons Within the Vagus Nerve" by Varun Malik

Thank you for submitting your manuscript to The Journal of Physiology. It has been assessed by a Reviewing Editor and by 1 expert referee and we are pleased to tell you that it is acceptable for publication following satisfactory revision.

The review comments are copied at the end of this email.

Please address all the points raised and incorporate all requested revisions or explain in your Response to Referees why a change has not been made. We hope you will find the comments helpful and that you will be able to return your revised manuscript within 2 weeks. If you require longer than this, please contact journal staff: jp@physoc.org.

REVISION CHECKLIST:

We look forward to receiving your revised submission.

Yours sincerely,

Harold Schultz
Senior Editor
The Journal of Physiology

EDITOR COMMENTS

Reviewing Editor:

Would replace the heart image with a simpler schematic. This will make the figure much more elegant.

We can try to get an artist to help. [Note from Editorial Office: please contact us if you need assistance with figures]

Senior Editor:

Thank you for submission of your perspective article to the Journal of Physiology for consideration. The article has been reviewed by authors of the focus article and found to be acceptable for publication pending minor suggestions/concerns raised. Please address the comments from the referee. It was suggested to the editors that the figure, although nicely summarising the current work and proposing important future directions, may benefit from some streamlining (and better explanation of the flow of the diagram in the figure legend) to improve clarity of points being made. The Journal can provide artistic assistance. Also, please address the list of requirements for publication in the journal included in this letter.

REFEREE COMMENTS

Referee #1:

Many thanks to Dr. Malik for undertaking this perspective piece on our recently accepted work. We are excited about this work and it is very gratifying to read of the enthusiasm of others for it. I have just a few comments on the piece, most of which concern fairly subtle points of accuracy.

Line 26

The description of the means by which we assessed cardiac and respiratory rhythmicity is not quite right, and we apologise if our description in the paper was not clear. The perspective piece says "Cardiac rhythmicity was determined by correlating nerve firing frequency with R wave frequency (ECG). Likewise, respiratory rhythmicity, with respiration rate.", suggesting that we simply plotted the nerve firing frequency against the heart rate and the respiratory rate, respectively. It would be more accurate to say that we looked for fluctuations in spike firing frequency that were phasic with the cardiac and respiratory cycles by constructing cross-correlograms.

Line 30

"Positive spikes with cardiac and respiratory rhythmicity were considered to resemble cardioinhibitory efferent neurons."

Only those neurones that were cardiac rhythmic and showed expiratory rhythmicity specifically (rather than respiratory rhythmicity) were classified as cardioinhibitory.

Line 40

"Of 15 participants, 9 had single unit recordings".

It would be slightly more accurate to say that single units were isolated from the recordings of 9 of the 15 participants.

Line 48

"Negative spikes (possible cardiac efferent neurons) also had higher frequency with lower heart rates, supporting their classification as cardiac efferent neurons."

In the paper, we are quite careful to state that there is currently no evidence from animal studies that unmyelinated axons with this firing pattern are cardioinhibitory. Rather, we point out that cardioinhibitory neurones with unmyelinated axons have been observed in the rat, as have (in separate studies) neurones with unmyelinated axons in the cardiac vagal branch that fire with cardiac and expiratory rhythmicity. Whether the two are one and the same is, we believe, unknown.

Line 51

"There are some important limitations."

We went to great lengths to discuss these limitations in the paper. May we humbly request that this is acknowledged?

Throughout

Sometimes the word 'nerve' is used when 'neuron(e)' or 'axon' would be more appropriate.

Many thanks again to Dr. Malik for undertaking this perspective piece.

END OF COMMENTS

Confidential Review

12-Aug-2024

Thank you for the opportunity to provide a perspective on this manuscript. I have made the suggested minor revisions to this perspectives piece as detailed below.

EDITOR COMMENTS

Reviewing Editor:

Would replace the heart image with a simpler schematic. This will make the figure much more elegant.

We can try to get an artist to help. [Note from Editorial Office: please contact us if you need assistance with figures]

Senior Editor:

Thank you for submission of your perspective article to the Journal of Physiology for consideration. The article has been reviewed by authors of the focus article and found to be acceptable for publication pending minor suggestions/concerns raised. Please address the comments from the referee. It was suggested to the editors that the figure, although nicely summarising the current work and proposing important future directions, may benefit from some streamlining (and better explanation of the flow of the diagram in the figure legend) to improve clarity of points being made. The Journal can provide artistic assistance. Also, please address the list of requirements for publication in the journal included in this letter.

I have replaced the anatomical heart sketch with the simplest outline of the heart. I have also made attempts to improve readability of the figure and have re-written the legend to help with this. I would be more than happy, if the editors feel, to make use of a journal artist to improvise further as required.

REFeree COMMENTS

Referee #1:

Many thanks to Dr. Malik for undertaking this perspective piece on our recently accepted work. We are excited about this work and it is very gratifying to read of the enthusiasm of others for it. I have just a few comments on the piece, most of which concern fairly subtle points of accuracy.

Line 26

The description of the means by which we assessed cardiac and respiratory rhythmicity is not quite right, and we apologise if our description in the paper was not clear. The perspective piece says "Cardiac rhythmicity was determined by correlating nerve firing frequency with R wave frequency (ECG). Likewise, respiratory rhythmicity, with respiration rate.", suggesting that we simply plotted the nerve firing frequency against the heart rate and the respiratory rate, respectively. It would be

more accurate to say that we looked for fluctuations in spike firing frequency that were phasic with the cardiac and respiratory cycles by constructing cross-correlograms.

Thank you for clarifying this point. I have amended the piece to reflect this methodology.

Line 30

"Positive spikes with cardiac and respiratory rhythmicity were considered to resemble cardioinhibitory efferent neurons."

Only those neurones that were cardiac rhythmic and showed expiratory rhythmicity specifically (rather than respiratory rhythmicity) were classified as cardioinhibitory.

This has also been amended.

Line 40

"Of 15 participants, 9 had single unit recordings".

It would be slightly more accurate to say that single units were isolated from the recordings of 9 of the 15 participants.

Accepted and revised.

Line 48

"Negative spikes (possible cardiac efferent neurons) also had higher frequency with lower heart rates, supporting their classification as cardiac efferent neurons."

In the paper, we are quite careful to state that there is currently no evidence from animal studies that unmyelinated axons with this firing pattern are cardioinhibitory. Rather, we point out that cardioinhibitory neurones with unmyelinated axons have been observed in the rat, as have (in separate studies) neurones with unmyelinated axons in the cardiac vagal branch that fire with cardiac and expiratory rhythmicity. Whether the two are one and the same is, we believe, unknown.

I do not believe that the statement in the perspective piece infringes upon the concepts that a. the classification of cardiac efferent neurons is not well-established (and this is a discussion point given the power of this important technique to be able to provide some discrimination in human physiology) and b. that it is unknown whether they are cardioinhibitory or not. This point is made to emphasise that the higher frequencies at lower heart rates are supportive of the classification that you have utilised to indicate cardiac specificity (rather than any intimation to suggest that they are cardioinhibitory). These are also discussed briefly later in the piece which clarifies this. I say simply that they this finding is "supportive of their involvement in cardiac control" rather than that they are cardioinhibitory. Such details remain important and can be viewed in the peer-reviewed manuscript. The perspective piece is far too short and must remain a (brief) summary of the work.

Line 51

"There are some important limitations."

We went to great lengths to discuss these limitations in the paper. May we humbly request that this is acknowledged?

I can unreservedly (whole-heartedly) agree that the authors' manuscript, together with considered and careful peer-review has resulted in a paper of high quality. This is evidenced by the decision to support a perspectives piece. I would also acknowledge here that it is especially beneficial to have discourse from the authors of the manuscript themselves to ensure that their work is accurately portrayed in this perspectives piece! There are limitations in every scientific work –this study is not exempt, and it is important for any perspectives piece to reflect on these limitations to provide a balanced overview of the work. Here, the mention of important limitations should not be taken to imply that these were not discussed in the paper, simply that there are limitations. Further, limitations form a critical part of the discussion of any paper. I have added a statement to this effect as I concur with the authorship that the current manuscript has a good limitations section.

Throughout

Sometimes the word 'nerve' is used when 'neuron(e)' or 'axon' would be more appropriate.

This is a fair statement. I have revised usage as is appropriate.

Many thanks again to Dr. Malik for undertaking this perspective piece.

It has been an honour to write an opinion piece on these important and exciting findings.

Dear Dr Malik,

Re: JP-P-2024-287381R1 "Direction From The Wanderer: Insight Into Cardiac Neural Control From Single Axons Within the Vagus Nerve" by Varun Malik

Thank you for submitting your manuscript to The Journal of Physiology. It has been assessed by a Reviewing Editor and by 1 expert referee and we are pleased to tell you that it is acceptable for publication following satisfactory minor revision.

The review reports are copied at the end of this email.

REVISION CHECKLIST:

We look forward to receiving your revised submission.

Yours sincerely,

Harold Schultz
Senior Editor
The Journal of Physiology

REQUIRED ITEMS

Please make sure you properly reference the Farmer et al. paper (i.e. give date in text citation, and add to References section).

EDITOR COMMENTS

Reviewing Editor:

Please make the final edits.

Please make sure you properly reference the Farmer et al. paper (i.e. give date in text citation, and add to References section).

Senior Editor:

Thank you for submission of your revised perspective article to the Journal of Physiology. The figure and legend are much improved. There was one remaining minor suggestion for the figure. I hate to further delay acceptance, but the suggestion for a figure can not be fixed in production. I might suggest in the figure: Veno-atrial junctions:right and left atria. No further revision of the text is required.

REFEREE COMMENTS

Referee #1:

Many thanks to the author for addressing or, where appropriate, rebutting our comments. Regarding line 48, they are quite right in what they say.

The figure is much clearer as revised. My only further suggestion would be to point out that, in addition to the left atrial receptors specified in the figure, there also exists a population of right atrial receptors (see below refs).

Kappagoda CT, Linden RJ & Mary DA (1976). Atrial receptors in the cat. J Physiol 262, 431-446.

Kappagoda CT, Linden RJ & Mary DA (1977). Atrial receptors in the dog and rabbit. J Physiol 272, 799-815.

Many thanks again to Prof. Malik.

END OF COMMENTS

1st Confidential Review

27-Aug-2024

Thank you for the opportunity to provide a perspective on this manuscript. I have made the required revisions.

REQUIRED ITEMS

Please make sure you properly reference the Farmer et al. paper (i.e. give date in text citation, and add to References section).

This has been added.

EDITOR COMMENTS

Reviewing Editor:

Please make the final edits.

Please make sure you properly reference the Farmer et al. paper (i.e. give date in text citation, and add to References section).

This has been added with basic details. The journal can add page details/DOI etc as they become available.

Senior Editor:

Thank you for submission of your revised perspective article to the Journal of Physiology. The figure and legend are much improved. There was one remaining minor suggestion for the figure. I hate to further delay acceptance, but the suggestion for a figure can not be fixed in production. I might suggest in the figure: Veno-atrial junctions:right and left atria. No further revision of the text is required.

Thank you- this has been amended.

REFEREE COMMENTS

Referee #1:

Many thanks to the author for addressing or, where appropriate, rebutting our comments. Regarding line 48, they are quite right in what they say.

The figure is much clearer as revised. My only further suggestion would be to point out that, in addition to the left atrial receptors specified in the figure, there also exists

a population of right atrial receptors (see below refs).

Kappagoda CT, Linden RJ & Mary DA (1976). Atrial receptors in the cat. *J Physiol* 262, 431-446.

Kappagoda CT, Linden RJ & Mary DA (1977). Atrial receptors in the dog and rabbit. *J Physiol* 272, 799-815.

Yes, quite true – it is ofcourse well known that these receptors exist in all venoatrial junctions in the heart- though perhaps with the highest numbers in the PV-LA (as per the work of Coleridge). I have amended the figure as recommended, though, once again, this is a detailed point. The co-location of these receptors in the PV-LA, a highly arrhythmogenic area and target for therapeutic intervention in humans, is important to get across.

Many thanks again to Prof. Malik.

Dear Dr Malik,

Re: JP-P-2024-287381R2 "Direction From The Wanderer: Insight Into Cardiac Neural Control From Single Axons Within the Vagus Nerve" by Varun Malik

We are pleased to tell you that your paper has been accepted for publication in The Journal of Physiology.

Authors should note that it is too late at this point to offer corrections prior to proofing. Major corrections at proof stage, such as changes to figures, will be referred to the Editors for approval before they can be incorporated. Only minor changes, such as to style and consistency, should be made at proof stage. Changes that need to be made after proof stage will usually require a formal correction notice.

If you would like to receive our 'Research Roundup', a monthly newsletter highlighting the cutting-edge research published in The Physiological Society's family of journals (The Journal of Physiology, Experimental Physiology and Physiological Reports), please click this link, fill in your name and email address and select 'Research Roundup': <https://www.physoc.org/journals-and-media/membernews/>

Yours sincerely,

Harold Schultz
Senior Editor
The Journal of Physiology

P.S. - You can help your research get the attention it deserves! Check out Wiley's free Promotion Guide for best-practice recommendations for promoting your work at www.wileyauthors.com/eeo/guide. You can learn more about Wiley Editing Services which offers professional video, design, and writing services to create shareable video abstracts, infographics, conference posters, lay summaries, and research news stories for your research at www.wileyauthors.com/eeo/promotion.

IMPORTANT NOTICE ABOUT OPEN ACCESS: To assist authors whose funding agencies mandate public access to published research findings sooner than 12 months after publication, The Journal of Physiology allows authors to pay an Open Access (OA) fee to have their papers made freely available immediately on publication.

You can check if your funder or institution has a Wiley Open Access Account here: <https://authorservices.wiley.com/author-resources/Journal-Authors/licensing-and-open-access/open-access/author-compliance-tool.html>.

EDITOR COMMENTS

The editors wish to thank the author for these final adjustments to the manuscript. The article is now accepted for publication. Congratulations for an interesting and insightful perspective on the focus article "FIRING PROPERTIES OF SINGLE AXONS WITH CARDIAC-RHYTHMICITY IN THE HUMAN CERVICAL VAGUS NERVE". Please consider the Journal of Physiology for your future works.